

# Are invasive populations characterized by a broader diet than native populations?

Julien Courant[1], Solveig Vogt[2,3], Raquel Marques[4], John Measey[3], Jean Secondi[5,6], Rui Rebelo[4], André De Villiers[3], Flora Ihlow[2], Charlotte De Busschere[7], Thierry Backeljau[7,8], Dennis Rödder[2] and Anthony Herrel[1,9]

[1] UMR 7179, Département d'Ecologie et de Gestion de la Biodiversité, Centre National de la Recherche Scientifique, Paris, France
[2] Herpetology Section, Zoologisches Forschungsmuseum Alexander Koenig, Bonn, Germany
[3] Centre for Invasion Biology, Stellenbosch University, Stellenbosch, South Africa
[4] Centre for Ecology, Evolution and Environmental Changes, Faculdade de Ciências da Universidade de Lisboa, Lisboa, Portugal
[5] UMR5023 Ecologie des Hydrosystèmes Naturels et Anthropisés, ENTPE, CNRS, Université de Lyon, Université Lyon 1, Villeurbanne, France
[6] UMR 6554 LETG –LEESA, Université d'Angers, Angers, France
[7] Royal Belgian Institute of Natural Sciences, Brussels, Belgium
[8] Evolutionary Ecology Group, University of Antwerp, Antwerp, Belgium
[9] Evolutionary Morphology of Vertebrates, Ghent University, Ghent, Belgium

Corresponding author
Julien Courant,
julien.courant@edu.mnhn.fr,
jcourant@hotmail.fr

## ABSTRACT

**Background**. Invasive species are among the most significant threats to biodiversity. The diet of invasive animal populations is a crucial factor that must be considered in the context of biological invasions. A broad dietary spectrum is a frequently cited characteristic of invasive species, allowing them to thrive in a wide range of environments. Therefore, empirical studies comparing diet in invasive and native populations are necessary to understand dietary requirements, dietary flexibility, and the associated impacts of invasive species.

**Methods**. In this study, we compared the diet of populations of the African clawed frog, *Xenopus laevis* in its native range, with several areas where it has become invasive. Each prey category detected in stomach contents was assigned to an ecological category, allowing a comparison of the diversity of ecological traits among the prey items in the diet of native and introduced populations. The comparison of diets was also performed using evenness as a niche breadth index on all sampled populations, and electivity as a prey selection index for three out of the six sampled populations.

**Results**. Our results showed that diet breadth could be either narrow or broad in invasive populations. According to diet and prey availability, zooplankton was strongly preferred in most cases. In lotic environments, zooplankton was replaced by benthic preys, such as ephemeropteran larvae.

**Discussion**. The relative proportions of prey with different ecological traits, and dietary variability within and between areas of occurrence, suggest that *X. laevis* is a generalist predator in both native and invasive populations. Shifts in the realized trophic niche are observed, and appear related to resource availability. *Xenopus laevis* may strongly impact aquatic ecosystems because of its near complete aquatic lifestyle and its significant consumption of key taxa for the trophic relationships in ponds.

## INTRODUCTION

Invasive species usually occupy a wide geographical range in their native area. Invasive species are typically characterized by a number of traits that favor the establishment and spread across new ecosystems, including a broad environmental tolerances, high genetic variability, rapid growth, early sexual maturity combined with a high reproductive rate, short generation time, broad diet, gregariousness, rapid dispersal, and they are often commensal (*Ricciardi & Rasmussen, 1998*). Of course, not all invasive species meet all these criteria (*Lodge, 1993*). For example, successful invaders do not necessarily exhibit a broad diet (*Vazquez, 2006*). Yet a large dietary niche breadth is frequently considered as a hallmark of an invasive taxon.

The dietary niche is a component of the Eltonian niche, defined as the position of an organism, exhibiting a null population growth rate, in the trophic relationships with others organisms of the ecosystem such as its nutrients, predators and competitors (*Chase & Leibold, 2003*). Another aspect of the ecological niche is the Grinnellian niche, defined as the set of all values of the abiotic parameters enabling the occupancy of an area by a species (*Soberón, 2007*). The Grinnellian niche has known a recent intensification of research with the development of species distribution models (e.g., *Angetter, Lötters & Rödder, 2011*; *Guisan et al., 2013*). Based on a set of occurrence records and predictor variables, these models determine a given species fundamental niche and facilitate the assessment of potential niche shifts when projected onto novel conditions. While these models are based on the assumption that species retain their ancestral traits over time (see *Ackerly, 2003*; *Wiens & Graham, 2005*) recent evidence (*Broennimann et al., 2007*; *Mukherjee et al., 2012*; *Stiels et al., 2015*) revealing shifts in realized Grinnellian niches on a macro-ecological scale call this concept into question. How far Eltonian niches operating on a population level are variable, under the assumption of niche conservatism, is less well studied. Whether a population maintains its characteristics, or shifts them in the course of the invasion process, will contribute determining its ecological impact. This information is therefore of crucial importance for conservation practitioners facing the threat posed by invasive species.

Native to southern Africa, the African clawed frog, *Xenopus laevis*, has been introduced in many countries on four continents, where accidentally and deliberately released individuals have established viable populations (see *Measey et al., 2012*). Despite its importance as a biological model organism (*Cannatella & De Sa, 1993*), and the abundance of invasive populations, few field studies have been undertaken in colonized ranges (for a review, see *Measey et al., 2012*). *Xenopus laevis* has been reported to negatively affect the invaded ecosystems, and as a consequence has been ranked as having the second greatest impact on native ecosystems by any amphibian (*Measey et al., 2016*). Shifts in the Grinnellian niche of *X. laevis* have been recently demonstrated (*Rödder et al., in press*), whereas studies on changes in the Eltonian niches have not yet been undertaken.

The diet of *X. laevis* has been studied in the species' native range of South-Africa (*Schoonbee, Prinsloo & Nxiweni, 1992*), as well as in several introduced populations in the United States of America (USA) (*McCoid & Fritts, 1980*), Wales (*Measey, 1998a*), Chile (*Lobos & Measey, 2002*), Italy (*Faraone et al., 2008*), Portugal (*Amaral & Rebelo, 2012*), and

France (*Courant et al., 2014*). In most studies, the majority of prey items are aquatic, with zooplankton and dipteran larvae being the most frequent. *Measey (1998b)* also noticed the importance of terrestrial prey. While stomach content analyses conducted in Portugal (*Amaral & Rebelo, 2012*) and in the USA (*McCoid & Fritts, 1980*) revealed that *X. laevis* consumed eggs of fishes and amphibians, no study has reported a direct impact linked to predation. A similar study, conducted in South African aquaculture ponds, revealed that farmed fish larvae constituted a large proportion (5–25% occurrence frequency according to fish size) of frog stomach contents (*Schramm, 1987*), while another study found *X. laevis* to consume large quantities of anuran eggs and larvae (*Vogt et al., 2017*).

Given the wide diversity of prey items, dietary studies on *X. laevis* usually suggest a generalist feeding behaviour, but only one study has thus far investigated prey electivity (*Measey, 1998a*). No previous studies have explicitly compared the feeding behavior of populations in different ecological contexts, and with different invasion histories. In this study, we compiled published data on the diet of *X. laevis* from the USA (*McCoid & Fritts, 1980*), Wales (*Measey, 1998a*), Chile (*Lobos & Measey, 2002*), and South Africa (*Vogt et al., 2017*) with data collected during recent field work in Portugal and France to test the hypotheses that (i) trophic niche breadth is wider in native populations, hence releaving the capacity of the species to readily adapt to novel environments; and (ii) the diet of invasive populations differs significantly between the invaded ranges depending on local prey availability, and thus resulting in a low degree of electivity and population-specific niche shifts.

## MATERIAL & METHODS

### Data sampling

Our dataset comprised 1,458 individuals from six countries, across four continents (Table 1). In most areas (Chile, South Africa, Wales, France), frogs were caught using funnel traps. In Portugal, animals were captured using electrofishing because the colonized habitats, mainly fast flowing streams, prevented the use of traps. In South Africa, research permission was issued by CapeNature (AAA007-01867) and SANParks (RC/2014-2015/001–2009/V1), with ethics clearance from Stellenbosch University Research Ethics Committee: Animal Care & Use (SU-ACUD15-00011). Animals from the Portuguese invasive population were captured under permit no 570/2014/CAPT from Instituto da Conservação da Natureza e das Florestas, in the scope of the ''Plano de erradicação de *Xenopus laevis* nas ribeiras do Concelho de Oeiras'' (Eradication plan of Xenopus laevis in the streams of Oeiras Municipality). In France, a research permit was provided by the prefecture of the Deux-Sèvres department.

Stomach content samples were obtained either by stomach flushing or dissection, following euthanasia of individuals by lethal injection of sodium pentobarbital or immersion in MS222. We considered that analyzing and comparing data collected with both dissection and flushing methods were valid and did not induce any bias (*Wu, Li & Wang, 2007*).

![PeerJ]

**Table 1  Characteristics of the methods used to capture and describe the diet of *Xenopus laevis*.**

|  | South Africa | Wales | France | Chile | Portugal | USA |
|---|---|---|---|---|---|---|
| **Population status** | Native | Extinct[a] | Invasive | Invasive | Invasive | Invasive |
| Period of capture | From 06/2014 To-09/2014 | From 05/1995 To 08/1996 | From 05/2014 To 10/2014 | 01/1998 03/2001 | From 06/2014 To 08/2014 | 1975–1976 |
| Geographical coordinates |  |  |  |  |  |  |
| Latitude/Longitude NW[b] | S34°18′24″ E18°25′35″ | N51°27′33″ W3°33′11″ | N47°16′14″ W0°33′56″ | S33°29′ W70°54′ | N38°45′09″ W9°17′27″ | See *McCoid & Fritts (1980)* |
| Latitude/Longitude SE[b] | S34°20′06″ E19°04′29″ | NA | N46°53′41″ W0°31′11″ | S33°37′ W70°39′ | N38°42′35″ W9°16′25″ | See *McCoid & Fritts (1980)* |
| Sampling design |  |  |  |  |  |  |
| Method | Trap | Trap | Trap | Trap | Electrofishing | NA |
| Capture occasion/site | From 1 to 4 | 29 | 3 | 1 | From 1 to 4 | 1 |
| Number of sites | 8 | 1 | 26 | 2 | 12 | 1 |
| Number of individuals | 164 | 375 | 438 | 48 | 352 | 81 |
| Prey availability | Yes | Yes | Yes | No | No | No |
| Habitat type | Ponds | Pond | Ponds | Ponds | Streams | Streams |
| Prey collection method | Flushing/ Dissection | Flushing | Dissection | Dissection | Dissection | Dissection |
| Published data |  |  |  |  |  |  |
| Prey frequency in stomachs | Yes | Yes | No | Yes | No | Yes |
| Niche breadth | Yes | No | No | No | No | No |
| Electivity | Yes | Yes | No | No | No | No |
| Individual data |  |  |  |  |  |  |
|  | Yes | No | Yes | Yes | Yes | No |

**Notes.**
[a]The population introduced in Wales went extinct twenty years after the data collection used in our study (*Tinsley et al., 2015*).
[b]Geographical coordinates (WGS 84), northwestern (NW) and southeastern corners (SE), of the minimum rectangle encompassing all sampled sites for $N_s > 1$.

## Data analysis

Prey items retrieved from the stomach content samples were identified to the lowest taxonomic level possible. However, for analytical consistency, we retained the lowest common taxonomic level that could be identified for all prey items. As volume and mass of prey items were not available for most studies, analyses were performed using prey frequencies. Each taxonomic prey category was assigned to one of the following ecological traits: plankton, benthos, nekton and terrestrial. Some groups of invertebrates belong to different ecological trait categories depending on their life stage, e.g., aquatic in their larval stage and terrestrial in their adult stage (Diptera, Ephemeroptera, Trichoptera). Thus, adults and larvae were treated separately when assigned to different ecological traits even though they belong to the same taxonomic prey category.

The diet of populations was first compared by calculating the relative abundance of each prey category and ecological trait. The relative abundance of prey classes (aquatic invertebrates, terrestrial invertebrates, vertebrates) was also calculated. To assess variation in diet between populations, a Principal Component Analysis (PCA) was performed on reduced and centered relative abundances of each prey category.

The concept of niche breadth can be applied to comparative studies of diet between populations or species (*Slatyer, Hirst & Sexton, 2013*), even if it has not always been treated within this contextual vocabulary (e.g., *Rehage, Barnett & Sih, 2005*; *Luiselli et al., 2007*; *Dalpadado & Mowbray, 2013*). We calculated niche breadth for all populations using the evenness measure J'. This index is based on the Shannon–Wiener's index H' (*Shannon & Weaver, 1964*), as recommended by *Colwell & Futuyma (1971)*:

$$J' = \frac{-\sum (p_i * \log p_i)}{\log n}$$

where $p_i$ is the proportion of the prey category $i$ in the diet and $n$ is the number of food categories.

To test whether J' is affected by the number of study sites the relationship of both variables was assessed using a nonlinear regression.

Prey availability was quantified in habitats for three of the six populations (France, Wales and South Africa) included in this study. Following *Measey (1998a)*, the same sampling method was applied to all populations. Prey selection was assessed using the *Vanderploeg & Scavia*'s (*1979*) relativized electivity index recommended by *Lechowicz (1982)*:

$$E^* = \frac{W_i - \frac{1}{n}}{W_i + \frac{1}{n}} \text{ with } W_i = \frac{\frac{r_i}{p_i}}{\sum r_i / p_i}$$

where $r_i$ is the relative abundance of prey category $i$ in the diet and $p_i$ is the relative abundance of prey category $i$ in the environment. The number of prey categories included in the analysis is represented by $n$.

## RESULTS

Across all samples zooplankton was the most common prey type with a mean relative abundance of 56.21% (Standard Deviation = 32.80%), followed by ephemeropteran larvae (10.31% ± 23.40%), dipteran larvae (9.68% ± 7.23%), and gastropods (7.24% ± 6.07%). The fifth and sixth most represented prey items were amphibian eggs, excluding *X. laevis* (4.86% ± 11.08%) and *X. laevis* eggs (3.46% ± 7.55%) respectively. Three aquatic invertebrate orders (Coleoptera, Diptera, and Heteroptera) were detected in all study sites (Table 2), while most of the terrestrial categories were exclusively found in one or two sites. Cannibalism of larvae and/or eggs was recorded in every locality, except Chile.

Aquatic invertebrates represented the most consumed prey item class, with a relative abundance ranging from 66% in South Africa to 99% in Wales. Terrestrial invertebrates were rarely consumed and consequently relative abundance ranged from 0.02% in France and the USA to 1.5% in Chile. Variability in relative abundance was highest for vertebrate prey, reaching a maximal relative abundance of 33% in South Africa while being absent in Chile.

According to the PCA (Fig. 1), the diet of the Portuguese and South African populations was respectively characterized by high relative abundances of ephemeropteran larvae, and eggs of native amphibians. The cluster representing Wales and Chile was characterized by a high occurrence of zooplankton and a very low occurrence of all vertebrate prey
**Table 2 Relative abundance (in %) of the prey items identified in the native and colonized ranges of *Xenopus laevis*.** When prey items are observed in very low quantities ($N < 3$), they are noted as <0.01% in the table. The main prey categories of each populations are underlined. Below the name of each ecological category, we mention the mean and the standard deviation of the relative abundance of items found in stomach contents.

| | Cat no | | South Africa | Wales | France | Chile | Portugal | USA |
|---|---|---|---|---|---|---|---|---|
| **Aquatic invertebrates** | | | **65.99** | **99.15** | **80.68** | **98.54** | **96.56** | **98.13** |
| | 1 | Zooplankton | 51.87 | 92.55 | 39.62 | 82.51 | 2.28 | 68.89 |
| **Benthos** | | | **8.86** | **6.45** | **37.08** | **14.85** | **93.90** | **27.41** |
| **34.80% ± 34.17%** | | | | | | | | |
| | 2 | Annelida | 0.03 | 0.00 | 0.02 | 0.00 | 0.05 | 0.00 |
| | 3 | Turbellaria | 0.00 | <0.01 | 0.00 | 0.00 | 0.00 | 0.00 |
| | 4 | Gastropoda | 0.00 | 0.00 | 9.82 | 10.64 | 15.05 | 7.99 |
| | 5 | Bivalvia | 0.00 | 0.24 | 0.44 | 0.00 | 0.05 | 0.00 |
| | 6 | Acari | 4.63 | 0.00 | 0.00 | 0.00 | 0.05 | 0.00 |
| | 8 | Amphipoda | 1.83 | 0.02 | 3.09 | 0.00 | 0.05 | 4.21 |
| | 9 | Isopoda | 0.00 | 0.00 | 0.00 | 0.00 | 2.62 | 0.05 |
| | 10 | Decapoda | 0.00 | 0.00 | 0.00 | 0.00 | 0.06 | 0.00 |
| | 11 | Diptera (larvae) | 1.78 | 6.14 | 20.90 | 4.21 | 15.24 | 9.91 |
| | 12 | Ephemeroptera (larvae) | 0.00 | 0.05 | 2.79 | 0.00 | 58.02 | 0.98 |
| | 13 | Trichoptera (larvae) | 0.59 | 0.00 | 0.02 | 0.00 | 2.71 | 4.27 |
| **Nekton** | | | **5.26** | **0.14** | **4.00** | **1.19** | **0.50** | **1.83** |
| **7.74% ± 13.87%** | | | | | | | | |
| | 14 | Coleoptera (larvae) | 1.97 | 0.03 | 0.02 | 0.00 | 0.06 | 0.27 |
| | 15 | Coleoptera (adult) | 0.67 | 0.09 | 1.74 | 0.44 | 0.38 | 0.34 |
| | 16 | Heteroptera | 1.09 | 0.02 | 1.12 | 0.18 | 0.01 | 0.16 |
| | 17 | Zygoptera (larvae) | 0.79 | 0.00 | 0.38 | 0.44 | 0.00 | 0.53 |
| | 18 | Anisoptera (larvae) | 0.74 | 0.00 | 0.74 | 0.13 | 0.05 | 0.53 |
| **Terrestrial invertebrates** | | | **0.86** | **0.43** | **0.23** | **1.46** | **0.70** | **0.24** |
| **0.64% ± 0.46%** | | | | | | | | |
| | 19 | Arachnida | 0.46 | 0.01 | 0.00 | 1.21 | 0.05 | 0.11 |
| | 20 | Isopoda | 0.00 | 0.05 | 0.00 | 0.03 | 0.04 | 0.00 |
| | 21 | Chilopoda | 0.00 | 0.00 | 0.00 | 0.00 | 0.01 | 0.00 |
| | 22 | Diplopoda | 0.00 | 0.00 | 0.00 | 0.00 | 0.01 | 0.00 |
| | 23 | Diptera | 0.09 | 0.06 | 0.00 | 0.00 | 0.04 | 0.08 |
| | 24 | Neuroptera | 0.04 | 0.00 | 0.00 | 0.00 | 0.00 | 0.00 |
| | 25 | Hymenoptera | 0.27 | 0.00 | 0.00 | 0.05 | 0.16 | 0.00 |
| | 26 | Coleoptera | 0.00 | 0.01 | 0.05 | 0.00 | 0.00 | 0.05 |
| | 27 | Lepidoptera (larvae) | 0.00 | <0.01 | 0.00 | 0.00 | 0.00 | 0.00 |
| | 28 | Lepidoptera (adult) | 0.00 | 0.00 | 0.00 | 0.15 | 0.00 | 0.00 |
| | 29 | Dermaptera | 0.00 | 0.01 | 0.00 | 0.03 | 0.00 | 0.00 |
| | 30 | Heteroptera | 0.00 | 0.00 | 0.02 | 0.00 | 0.10 | 0.00 |
| | 31 | Annelida | 0.00 | 0.02 | 0.09 | 0.00 | 0.02 | 0.00 |
| | 32 | Orthoptera | 0.00 | 0.00 | 0.08 | 0.00 | 0.00 | 0.00 |
| | 33 | Aphids | 0.00 | 0.27 | 0.00 | 0.00 | 0.00 | 0.00 |

**Table 2** (*continued*)

| | Cat no | | South Africa | Wales | France | Chile | Portugal | USA |
|---|---|---|---|---|---|---|---|---|
| | 34 | Trichoptera | 0.00 | 0.00 | 0.00 | 0.00 | 0.02 | 0.00 |
| | 35 | Ephemeroptera | 0.00 | 0.00 | 0.00 | 0.00 | 0.24 | 0.00 |
| **Vertebrates** | | | **33.13** | **0.41** | **19.09** | **0.00** | **2.74** | **1.63** |
| 9.45% ± 13.54% | | | | | | | | |
| | 36 | Fish (adult) | 0.00 | 0.00 | 0.11 | 0.00 | 0.02 | 0.08 |
| | 37 | Fish (egg) | 0.00 | 0.00 | 0.00 | 0.00 | 0.82 | 0.00 |
| | 38 | Amphibia (adult) | 0.14 | 0.00 | 0.02 | 0.00 | 0.00 | 0.00 |
| | 3 | Amphibia (larvae) | 3.37 | 0.00 | 0.00 | 0.00 | 0.00 | 0.00 |
| | 39 | Amphibia (egg) | 27.62 | 0.00 | 0.08 | 0.00 | 1.65 | 0.00 |
| | 40 | *X. laevis* (larvae) | 0.04 | 0.00 | 0.08 | 0.00 | 0.01 | 0.03 |
| | 41 | *X. laevis* (egg) | 0.00 | 0.41 | 18.81 | 0.00 | 0.00 | 1.52 |
| | 42 | Amphibia (rest) | 1.98 | 0.01 | 0.00 | 0.00 | 0.22 | 0.00 |
| | 43 | Bird (feather) | 0.00 | <0.01 | 0.00 | 0.00 | <0.01 | 0.00 |
| | 44 | Mammals | 0.00 | <0.01 | 0.00 | 0.00 | 0.00 | 0.00 |

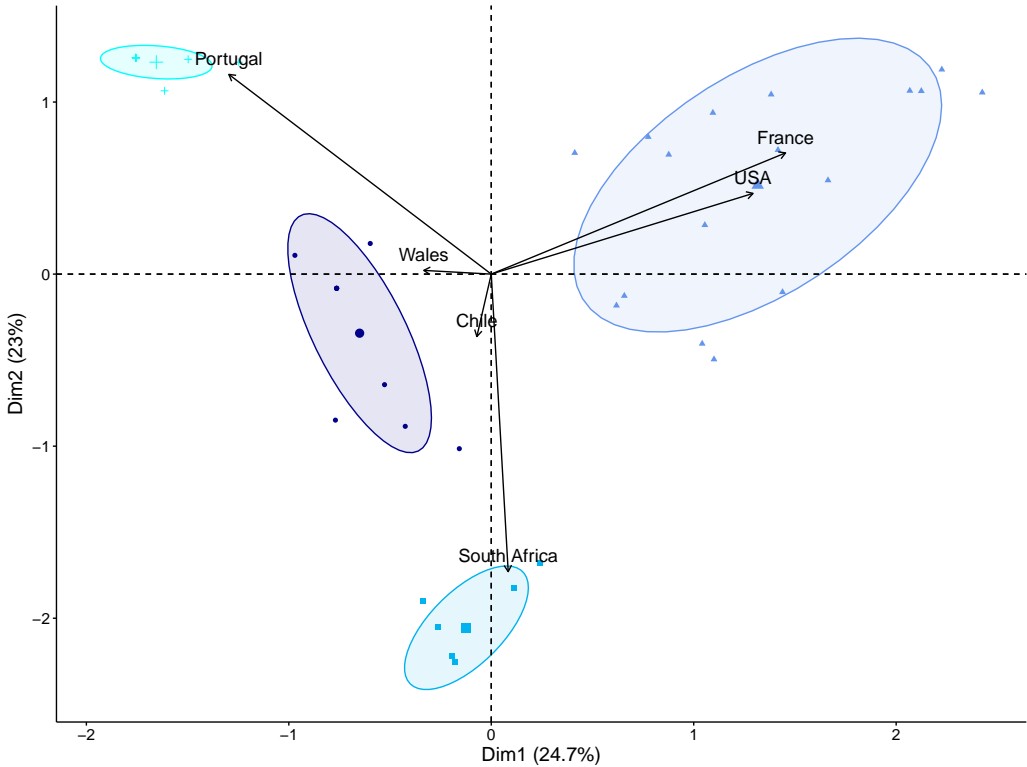

**Figure 1** Principal components of the diet of the native (South Africa) and invasive populations of *Xenopus laevis*, with prey categories as individuals (dots, squares, triangles and crosses) and populations as variables (black arrows).

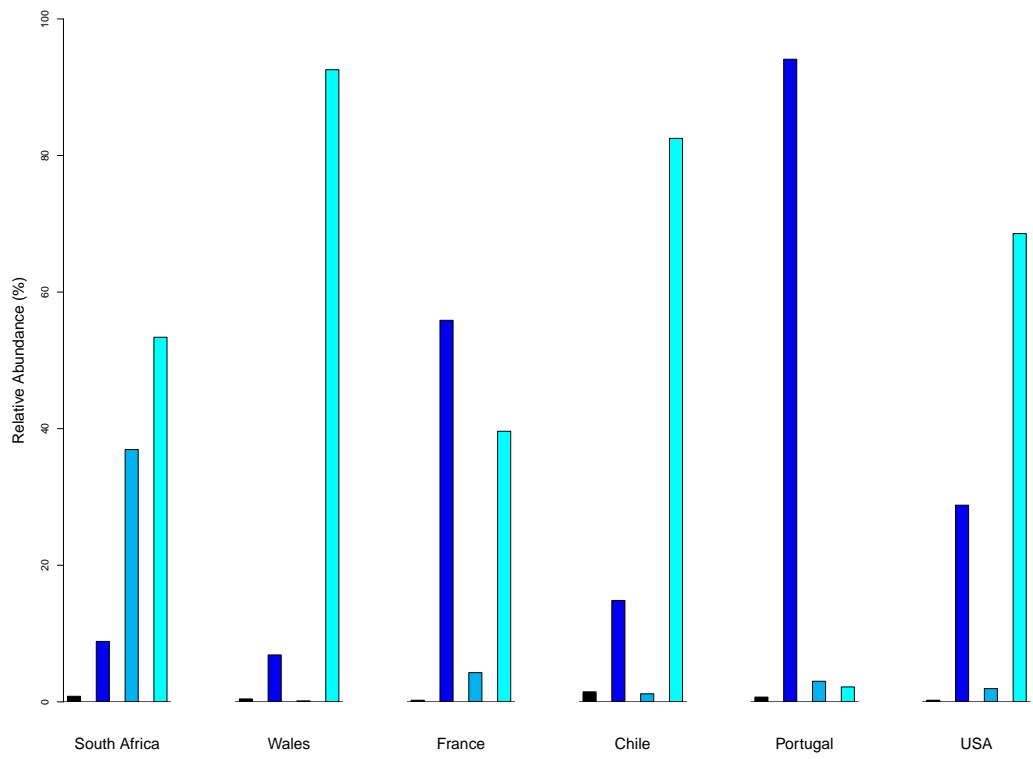

**Figure 2** **Occurrence of the main ecological traits among the native population (South Africa) and the five populations of *Xenopus laevis*.** Terrestrial prey (black), benthic prey (dark blue), nektonic items (sky blue) and planktonic prey (cyan).

items. Except for the eggs of *X. laevis,* French and American samples share the three most abundant prey categories (i.e., Diptera, Gastropoda, and Zooplankton), which explains the short distance between these samples in the PC space.

Benthic taxa represented 8.54% of the prey categories in South Africa, 6.87% in Wales, 14.85% in Chile, 28.8% in the USA, 55.87% in France, and 93.89% in Portugal (Fig. 2). Nektonic taxa represented 39.14% of the prey items in South Africa, 0.14% in Wales, 1.18% in Chile, 1.94% in the USA, 4.20% in France, and 3.01% in Portugal.

Standardized evenness was 0.48 in South Africa, 0.10 in Wales, 0.27 in Chile, 0.42 in the USA, 0.55 in France, 0.41 in Portugal (Fig. 3A). The relationship between evenness ($J'$) and the number of sampled sites ($N_s$) followed a logarithmic function ($J' = 0.086 * \ln(N_s) + 0.249$; Fig. 3B). The slope of the curve decreased from one to ten sites, and stabilized at 25 sites. There was no correlation between trophic niche breadth estimations and the number of stomach contents analyzed per population.

Negative electivity values were observed for most prey items (Fig. 4). In contrast, zooplankton was preferred (positive electivity) in the Welsh, South African and French populations. The electivity values of the other prey categories were variable, indicating that they were either selected, avoided, or not represented in the sampling.
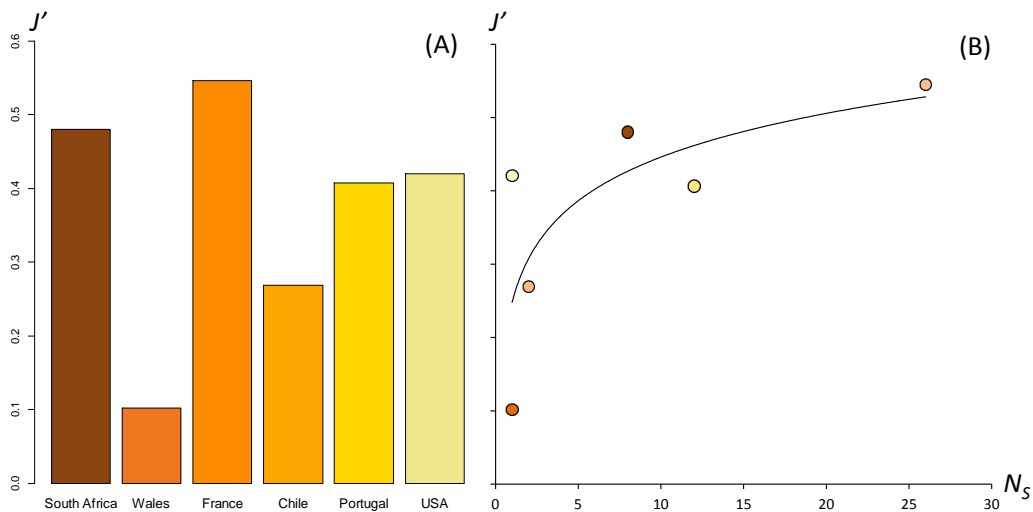

**Figure 3** **Niche breadth calculated for diet data in native and colonized ranges of *Xenopus laevis*.** Calculated with the Evenness $J'$ (A) and relationship between $J'$ and the number of sites $N_S$ used in localities (B).

# DISCUSSION

In studies focusing on invasive species, both niche conservatism and niche shifts are commonly reported in the literature (*Tillberg et al., 2007*; *Caut, Angulo & Courchamp, 2008*; *Comte, Cucherousset & Olden, 2016*). Here, we report strong modifications in the realized dietary niche between naturally occurring and introduced or invasive populations of *X. laevis*. While these differences mainly constituted contractions or expansions of the realized dietary niche, the diet of the Portuguese population represented a shift in the species' diet. Individuals from this population were captured in fast flowing streams using electrofishing, while all others frogs were caught in lentic environments. The difference might therefore be attributed to the habitat characteristics, reinforcing the hypothesis that *X. laevis* is a generalist predator that modifies its diet according to available resources.

Our study provides the first analysis of prey availability for a large number of sites that include both natural and invaded areas. Our findings indicate that *X. laevis* may expand or shift some dimensions of its trophic niche in novel environments. This result is significant, and has clear consequences for evaluating conditions that favor the invasion of newly introduced populations. Suitable trophic conditions allowing a positive population growth rate may be found in many places where prey abundance is sufficient. A recent macroecological assessment revealed that large areas of suitable climatic conditions were available outside the species' native range, and that *X. laevis* was likely to expand its range in Europe as a result of climate change (*Ihlow et al., 2016*). The broad global trophic niche of *X. laevis* and its ability to adapt its diet according to local conditions, contribute to the strong invasive potential of this species, and the high impact it may induce on its environment (see *Kumschick et al., 2017*).

The diet of *X. laevis* has been studied in its native range (*Schoonbee, Prinsloo & Nxiweni, 1992*; *Vogt et al., 2017*), and in different invasive populations (*McCoid & Fritts, 1980*;
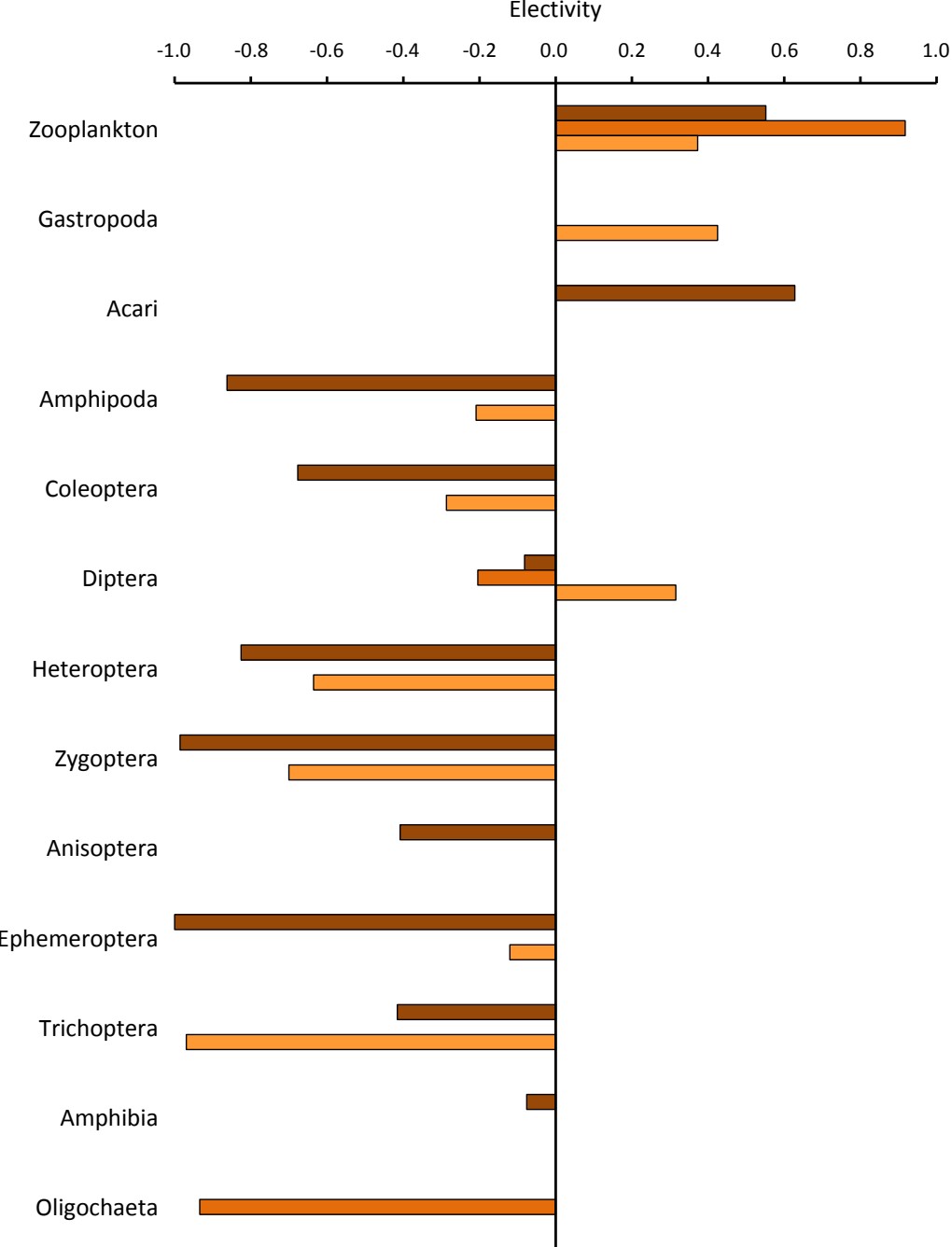

**Figure 4** Electivity index for each aquatic prey category consumed in the native population of *Xenopus laevis* in South Africa (brown) and the invasive populations of Wales (dark-orange) and France (light-orange).

*Measey, 1998a*; *Lobos & Measey, 2002*; *Faraone et al., 2008*; *Amaral & Rebelo, 2012*; *Courant et al., 2014*). The first study carried out in the native range was performed in a fish farm (*Schoonbee, Prinsloo & Nxiweni, 1992*), which does not necessarily represent the typical diet of native populations. In most studies, including ours, *X. laevis* was found to predominantly
consume relatively small prey items. In Portugal, *X. laevis* mostly inhabits streams where zooplankton is rare and the main prey items are benthic ephemeropteran larvae. Prey availability in streams was not studied in Portugal, but collecting such data may help understanding the dietary shift we observed in this population.

In all populations but Chile, predation on amphibians was observed, including *X. laevis* itself, which represented the most frequent vertebrate taxon in the collective sample. This cannibalistic behavior has been recorded in non-native populations, including in the USA (*McCoid & Fritts, 1980*), Wales (*Measey, 1998a*), Chile (*Lobos & Jaksic, 2005*), and Italy (*Faraone et al., 2008*) but the high frequencies observed in France and South Africa are unprecedented. In the Chilean population, autochthonous amphibians, as well as *X. laevis* eggs and larvae, were not observed during the sampling period (J Measey, pers. comm., 2016) which may explain their absence in the diet. The small number of stomach contents sampled and sites analyzed for this population may also provide a biased assessment of the putative absence of native amphibians in this area. Predation on amphibians was minor in most populations, except for the native range of *X. laevis*. This low occurrence of predation on amphibians in other localities may be related to the season during which studies were carried out, and possible changes in the behavior of native amphibians which have co-existed with *X. laevis* for decades. Our study does not provide any evidence supporting this idea, but ongoing studies should bring new insights into this question. The noteworthy anurophagy reported in this study corroborates the conclusions of *Measey et al. (2016)* regarding the occurrence of this behavior among pipids.

In this study, terrestrial invertebrates represented the least consumed prey class. However, cumulatively, there were as many taxonomic categories among terrestrial, as among aquatic invertebrates. The occurrence of terrestrial prey items did not vary much between populations, suggesting that there were no local specializations for the capture of terrestrial prey. In previous dietary studies of *X. laevis*, authors concluded that the high portion of terrestrial prey could not exclusively be explained by the capture of terrestrial invertebrates that had fallen into the water (*Measey, 1998b*).

From a methodological perspective, sampling effort in each locality was heterogeneous with respect to the sampling period, the number of prospected sites (range: 1–26), and the number of individuals analyzed per population. Consequently, niche breadth could be under-estimated in populations where only few sites were sampled (Wales, Chile, and USA) or where sites were inter-connected (streams in Portugal). According to our results, there is a positive relationship between diet breadth and the number of prospected sites. As no threshold was identified for an optimal number of study sites, we would recommend using as many spatial and temporal replicates as possible for studies aiming at comparing niche breadths. The diet of an individual may be influenced by its size (*Schafer et al., 2002*), age (*Gales, 1982*; *Rutz, Whittingham & Newton, 2006*), and sex independently of size (*Gales, 1982*; *Göçmen et al., 2011*; *Van Ngo, Lee & Ngo, 2014*). In our study, these data were not available for every population, preventing us from analyzing their effect on diet breadth. In other respects, the unique native population included in our study may not be representative of the diet in the native range. Estimations of niche breadth and prey selectivity may have differed had they been based on a larger sample of native populations.

Therefore, we encourage future investigators to consider as many naturally occurring populations as invasive populations in studies aimed at understanding the feeding ecology of invasive animals.

Our results indicate that no prey categories are strongly selected, except for zooplankton in Wales. This suggests that *X. laevis* does not usually specializes its diet and hence does not develop a population specific dietary niche. This characteristic may enhance its capacity to establish and spread in novel environments. Potential perturbations of *X. laevis* in its environment, linked to the large predation on small prey items, are crucial elements of the trophic relationships in aquatic ecosystems that still need to be demonstrated. Our results reflect the diet of *X. laevis* in invasive populations after decades of colonization and do not necessarily reflect the diet of the species at the moment of introduction. Some invasive species modify their diet during the years, or decades following habitat colonization (e.g., *Tillberg et al., 2007*; *Gkenas et al., 2016*). Comparing the diet of individuals at the core and the edge of a newly colonized area may be an effective approach to investigate the change in dietary composition of *X. laevis* during the colonization process.

## ACKNOWLEDGEMENTS

We are sincerely grateful to the two anonymous reviewers and the Academic Editor for their helpful and constructive comments on the manuscript.

### Funding

This research was funded by BiodivERsA BR/132/A1/INVAXEN-BE ''Invasive biology of *Xenopus* laevis in Europe: ecology, impact and predictive models'', the DST-NRF Centre of Excellence for Invasion Biology and the National Research Foundation of South Africa (NRF Grant No. 87759) to John Measey. The funders had no role in study design, data collection and analysis, decision to publish, or preparation of the manuscript.

### Grant Disclosures

The following grant information was disclosed by the authors:
DST-NRF Centre of Excellence for Invasion Biology and National Research Foundation of South Africa: 87759.

### Competing Interests

John Measey is an Academic Editor for PeerJ.

### Author Contributions

- Julien Courant conceived and designed the experiments, performed the experiments, analyzed the data, contributed reagents/materials/analysis tools, wrote the paper, prepared figures and/or tables, reviewed drafts of the paper.
- Solveig Vogt, Raquel Marques and André De Villiers performed the experiments, wrote the paper, reviewed drafts of the paper.

- John Measey conceived and designed the experiments, performed the experiments, contributed reagents/materials/analysis tools, wrote the paper, reviewed drafts of the paper.
- Jean Secondi and Anthony Herrel conceived and designed the experiments, contributed reagents/materials/analysis tools, wrote the paper, reviewed drafts of the paper.
- Rui Rebelo conceived and designed the experiments, performed the experiments, wrote the paper, reviewed drafts of the paper.
- Flora Ihlow, Charlotte De Busschere, Thierry Backeljau and Dennis Rödder wrote the paper, reviewed drafts of the paper.

## Animal Ethics

The following information was supplied relating to ethical approvals (i.e., approving body and any reference numbers):

In South Africa, research permission was issued by CapeNature (AAA007-01867) and SANParks, with ethics clearance from Stellenbosch University Research Ethics Committee: Animal Care & Use (SU-ACUD15-00011). Animals from the Portuguese invasive population were captured under permit no 570/2014/CAPT from Instituto da Conservação da Natureza e das Florestas, in the scope of the ''Plano de erradicação de *Xenopus laevis* nas ribeiras do Concelho de Oeiras ("Eradication plan of Xenopus laevis in the streams of Oeiras Municipality").'' In France, a research permit was provided by the prefecture of the Deux-Sèvres department.

## Data Availability

The raw data has been supplied as a Supplementary File.

## Supplemental Information

Supplemental information for this article can be found online at http://dx.doi.org/10.7717/peerj.3250#supplemental-information.

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
