# Peer review of "Are invasive populations characterized by a broader diet than native populations?"

_PeerJ, doi:10.7717/peerj.3250_

## Round 0.1 · original submission · Major Revisions

This is an interesting paper that is largely ok but could do with some reformulating as described by one of the reviewers. I believe this may take some time hence Major Corrections but I do not believe it will be particularly difficult to achieve.

Reviewer 1 ·

Basic reporting

The manuscript generally adopts a professional approach, and presents detailed lines of arguments which are well connected to the evidence presented. However, the manuscript also lacks attention to detail, and requires a further formal overhaul. It contains a fairly large number of rather awkward phrases and syntax error (e.g. the use of commas needs checking throughout). Below is an example of points from the Title, Abstract and Introduction which should become considered during a revision (I refrained from presenting such a list for the other chapters).

Title: Is the second “invasive” necessary?

Line 19: “The diet of a species is a crucial factor that must be considered….” Is not applicable to e.g. invasive plants. Either rephrase the sentence or explicitly state that the argument is based on invasive animals only.

Line 23: “….to understand the need of invasive species….” is awkward (this sentences doesn’t want to say that we need invasive species!). Rephrase to “..to understand the diet of invasive species.” or similar.

Line 27: “….allowing a comparison of the relative abundance of ecological traits in the diet of populations.” is again awkward. Ecological traits represented by prey might be reflected in the diet, but traits themselves are not consumed. Rephrase.

Lines 29&30: delete “the” before Evenness and Electivity

Lines 39&40: I found the phrase “intensive predation on key taxa” fishing for significance. Where is the evidence that predation was intense, and in what respect are the consumed organisms key species?

Line 51&52: “….the environmental niche concept has been studied, specifically in the context of……..” is another example for an awkward phrase.

Line 58 and 59: It is probably useful for many readers to explain the Grinellian and Eltonian niche concepts here.

Line 61&62: the phrase “landscape management decisions” appears too vague or out of context here. Is it invasive species management decisions?

Lines 62-64: the last sentence of this paragraph is not well connected to the reminder of arguments, as the reader is not yet informed about the study species yet – move e.g. to the next paragraph which introduces the clawed frog.

Lines 90-92: “…published data on the diet of X. laevis from……South Africa (Vogt et al., under review)” is a little misleading and potentially problematic. Firstly, the citation is missing in the reference list. Secondly, a manuscript under review is not published. Thirdly, that an as yet inaccessible paper is based on the same data makes it impossible to assess the validity of the present work as a standalone paper.

Experimental design

The study is based on an overall large sample size, and the data are overall well collected and analyzed. The study rationale is well described.

However,to justify the present manuscript as a standalone paper. more details are needed on the extent to which the presented data and analyses have been published previously (e.g. as footnotes in the Tables). It is fine to re-use existing data, but have e.g. the niche metrices been published before? As the manuscript stands, the amount of overlap of the present work with Vogt et al. under review is also impossible to assess, as this paper is not yet in the public domain.

As far as I can overlook it, more details such as research permits are also required for the "data collected during recent field work in Portugal and France" (line 92).

Validity of the findings

The main findings -albeit not earth shattering- are relevant and meaningful. However, a conceptual weakness of the study is that, while data from five locations across three continents are available for invasive populations, data from only a single location are available for native populations. The high degree of diet opportunism shown by the present work combined with a large geographic range questions whether this single location is a true representative of the species' native diet spectrum. While this is not questioning the validity or the study, the authors need to consider this weakness in more detail in e.g. the Discussion.

Additional comments

None required.

Reviewer 2 ·

Basic reporting

Well laid out, clear, logical.

Some attention needs to be made to punctuation and grammar.

Figures would benefit from some colour, if possible. Tables require reformatting for consistency and clarity.

Experimental design

Clearly stated questions and methodological approach.

Validity of the findings

I enjoyed reading this interesting study on an important area of invasive species biology. The spread, invasiveness, and dietary intake of X. laevis are factors that will become increasingly important in the face of global climate change, being as it is likely to facilitate range expansion and severity of species impact in naive systems.

The authors present a clear concise assessment of X. leavis diet in its home range, and parts of the species introduced range. There are clear potential impacts, with significant connotations for conservation biology.

Additional comments

Comments/suggested edits are in the attached document.

Annotated reviews are not available for download in order to protect the identity of reviewers who chose to remain anonymous.

---

## Round 0.2 · Minor Revisions

The reviewers and I are happy with the corrections you have made. One of the reviewers has picked up on some very minor points/typos. If you could make these corrections which should take very little time I would be happy to accept.

Reviewer 1 ·

Basic reporting

The paper underwent the required general formal overhaul. It has significantly improved since its initial submission, and is now ready for publication. The presentation is clear and professional throughout. I only found very minor formal points such as the ones below, all of which can be dealt with at the page proof stage if necessary:

line 161: avoid back-to-back brackets
line 184: move the reference to Fig 3b into the brackets
references, line 380: use italics for “Xenopus laevis”

Experimental design

The experiments are described in sufficient details, with all strenghts and weaknesses highlighted as appropriate

Validity of the findings

The data are well analyses, and all conclusions are backed up with the required evicence

Additional comments

None required

Reviewer 2 ·

Basic reporting

See comments below

Experimental design

See comments below

Validity of the findings

See comments below

Additional comments

The manuscript is much improved, and the authors have taken on board and addressed all comments/edits. A few tweaks here and there (see attached annotated pdf), and I look forward to seeing this paper published.

Annotated reviews are not available for download in order to protect the identity of reviewers who chose to remain anonymous.

---

## Round 0.3 · accepted · Accept

Many thanks for making these corrections so quickly and I look forward to seeing it in press.